# DLP-LoRA: Efficient Task-Specific LoRA Fusion with a Dynamic, Lightweight Plugin for Large Language Models

## Abstract

Recent advancements in Large Language Models (LLMs) have achieved robust performance across diverse tasks, but fine-tuning these models for specific domains remains resource-intensive. Parameter-Efficient Fine-Tuning (PEFT) methods like Low-Rank Adaptation (LoRA) address this challenge by fine-tuning a small subset of parameters. However, existing methods for fusing multiple LoRAs lack dynamic fusion based on contextual inputs and often increase inference time due to token-level operations. We propose DLP-LoRA, a Dynamic Lightweight Plugin that employs a mini-MLP module with only 5M parameters to dynamically fuse multiple LoRAs at the sentence level using top-$p$ sampling strategies. This approach reduces inference time to less than twice that of single LoRA inference by leveraging parallel computation. Evaluations across 26 tasks—including multiple-choice questions and question answering—demonstrate that DLP-LoRA achieves an average accuracy of 92.34% on multiple-choice datasets and significant improvements in BLEU and ROUGE scores on QA datasets, outperforming different LLMs backbones under composite task settings. DLP-LoRA effectively balances performance and efficiency, making it a practical solution for dynamic multi-task adaptation in LLMs.

## 1 Introduction

Recent advancements in Large Language Models (LLMs) such as LLaMA 3.1 (Dubey et al., 2024), Qwen 2.5 (Team, 2024), and Gemma 2 (Team et al., 2024) have led to robust and superior performance across multiple benchmarks (Muennighoff et al., 2022; Ilyas Moutawwakil, 2023; Fourrier et al., 2024). These models have demonstrated remarkable capabilities in diverse areas, including code generation (Bai et al., 2023), mathematical reasoning (Ahn et al., 2024), and question answering (Achiam et al., 2023). Despite these achievements, fine-tuning all parameters of such large models for specific domains remains resource-intensive and time-consuming.

Parameter-Efficient Fine-Tuning (PEFT) methods (Houlsby et al., 2019; Xu et al., 2023) address this challenge by enabling the fine-tuning of a small subset of parameters, thereby improving performance in various applications like multi-task learning (Xu et al., 2024; Kong et al., 2024), multilingual summarisation, and transfer learning (Whitehouse et al., 2024; Zhao et al., 2024). One prominent PEFT approach is Low-Rank Adaptation (LoRA) (Hu et al., 2021), which fine-tunes low-rank matrices to capture domain-specific knowledge and merges them with pre-trained LLMs.

To enhance the multi-task learning capabilities of LLMs, several methods have been proposed to fuse task-specific LoRAs, including MoLE (Wu et al., 2024b), S-LoRA (Sheng et al., 2023), and LoRAHub (Huang et al., 2023). These approaches primarily use learnable gating networks or automatic loading mechanisms to combine multiple LoRAs. For instance, MeteoRA (Xu et al., 2024) introduces a token-level gating network to all attention and MLP layers for dynamic LoRA fusion.

However, most of these methods lack the ability to dynamically fuse LoRAs based on contextual prompt inputs during inference. They either require manual selection before combining LoRAs or necessitate additional fine-tuning when tasks change. Moreover, existing LoRA mixture strategies like MeteoRA focus on token-level Mixture-of-Experts (MoE) gating across all attention heads and MLP layers, which significantly increases inference time for next-token generation. Observations

from prior studies (Xu et al., 2024; Lin et al., 2024b; Muqeeth et al., 2024) indicate that within the same sentence of a task, the same LoRA is consistently assigned to each token. This suggests that token-level LoRA MoE might be unnecessary and computationally inefficient.

In this paper, we propose a Dynamic Lightweight Plugin for LoRA fusion (DLP-LoRA), which employs a lightweight MLP module to dynamically fuse multiple LoRAs based on top-$p$ sampling strategies. This mini-MLP plugin, containing only 5M parameters, is fast to train for multi-task classification and easily adaptable to new domains. By leveraging sentence-level LoRA selection and fusion guided by the mini-MLP plugin, DLP-LoRA requires less than twice the inference time compared to manually selecting and loading a single LoRA, making it comparable in efficiency.

We evaluate DLP-LoRA across 26 tasks, including 17 multiple-choice question (MCQ) datasets spanning mathematical QA, logical reasoning, language identification, and reading comprehension, as well as 9 question-answering (QA) datasets focused on summarisation, machine translation, and open-domain QA. Under comparable inference times to single LoRA setups, DLP-LoRA achieves an average accuracy of **92.34%** across the 17 MCQ datasets and average BLEU, ROUGE-1, and ROUGE-L scores of **57.62**, **56.03**, and **53.96**, respectively, across the 9 QA datasets. These evalua­tions are conducted using Qwen-2 1.5B, Qwen-2 7B, LLaMA-2 7B, and LLaMA-3 8B backbones. Additionally, our model demonstrates relative improvements of **92.95%** and **13.2%** for the MCQ and QA tasks, respectively, compared to different LLM backbones under composite task settings. With DLP-LoRA, the inference speed and performance of the Qwen-2 1.5B backbone are improved by over **90.90%** and **82.55%** under composite-26 task setting, respectively, when compared to the baseline LLaMA-2 13B. Our case studies further illustrate that sentence-level DLP-LoRA effec­tively balances the trade-off between multi-LoRA inference and fusion.

In summary, our contributions are threefold:

- We introduce DLP-LoRA, a dynamic and lightweight plugin for multi-LoRA selection and fusion that is fast to train and easily adaptable to new domains.

- By employing sentence-level multi-LoRA selection and fusion, DLP-LoRA leverages par­allel CUDA acceleration, achieving less than twice the inference time compared to single LoRA inference and outperforming token-level MoE gating routers in efficiency.

- Through extensive evaluations on 26 tasks including MCQ and QA, DLP-LoRA achieves performance comparable to single-task LoRA models and significantly improves accuracy and ROUGE metrics under composite task settings.

## 2 BACKGROUND

**Low-Rank Adaption.** Low-Rank Adaptation (LoRA) (Hu et al., 2021) is a method developed to fine-tune large language models (LLMs) for specific downstream tasks with enhanced efficiency by minimising the number of trainable parameters. Instead of updating all the model's parameters during training, LoRA introduces supplementary low-rank matrices. In Transformer-based autore­gressive LLMs, this technique involves freezing the pre-trained weights and integrating trainable low-rank matrices into designated layers, thereby substantially reducing computational overhead. The primary motivation for LoRA stems from the recognition that many parameter updates dur­ing fine-tuning occur within a low-dimensional subspace, indicating that full-rank weight updates are often unnecessary. By employing low-rank approximations, LoRA significantly decreases the number of parameters required for training—sometimes by factors as large as 10,000—while still maintaining competitive performance levels.

Formally, consider a weight matrix $\boldsymbol{W} \in \mathbb{R}^{h \times d}$ within the original LLMs. LoRA introduces two low-rank matrices, $\boldsymbol{A} \in \mathbb{R}^{h \times r}$ and $\boldsymbol{B} \in \mathbb{R}^{r \times d}$, where $r \ll \min(h, d)$. Instead of directly updating the weight matrix, LoRA modifies the model's forward pass according to the following equation:

$$\boldsymbol{W}' = \boldsymbol{o} + \Delta\boldsymbol{o} = \boldsymbol{W} + \boldsymbol{A}\boldsymbol{B} \tag{1}$$

Here, $\boldsymbol{W}'$ represents the adjusted weight matrix, while $\boldsymbol{A}$ and $\boldsymbol{B}$ are the trainable matrices incorpo­rated by LoRA. Consequently, the forward computation for an input $\boldsymbol{x} \in \mathbb{R}^{1 \times d}$ is expressed as:

$$\boldsymbol{h} = \boldsymbol{x}\boldsymbol{W}' = \boldsymbol{x}(\boldsymbol{o} + \Delta\boldsymbol{o}) = \boldsymbol{x}(\boldsymbol{W} + \boldsymbol{A}\boldsymbol{B}) = \boldsymbol{x}\boldsymbol{W} + \boldsymbol{x}\boldsymbol{A}\boldsymbol{B} \tag{2}$$

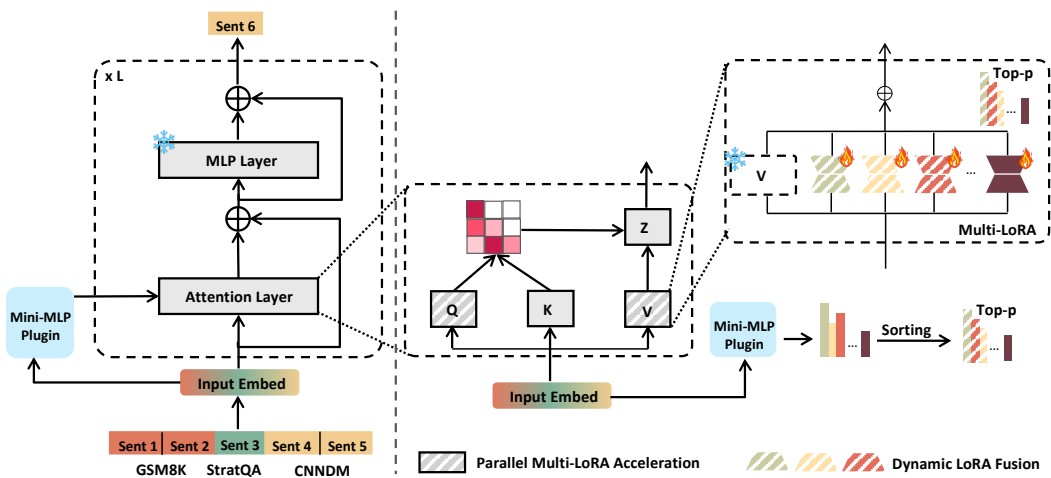

Figure 1: DLP-LoRA framework: different LoRAs will be activated based on the input task and sentence via mini-MLP plugin. When Top-$p$ sampling is used via the mini-MLP plugin, multiple LoRAs will be sampled and fused with probability $p$ as the threshold. DLP-LoRA fusion is only enabled once the first token of every new sentence is generated.

This approach guarantees that during the inference phase, after the training process is finalised, the low-rank matrices $A$ and $B$ can be integrated with the original weights $W$, thereby removing any additional computational overhead.

LoRA is predominantly applied to the attention projection matrices within the self-attention mechanisms of Transformer architectures, specifically targeting the query, key, and value projections, as well as the output projection. Recently, MLP layers can also be applied by LoRA (Dou et al., 2024; Li et al., 2024). This targeted application enhances the method's overall efficiency.

The minimalistic design of LoRA renders it especially beneficial for environments with limited computational resources or for applications necessitating the swift adaptation of extensive models. By keeping the majority of the model's parameters fixed and concentrating solely on learning the low-rank modifications, LoRA substantially decreases both memory usage and computational demands during the fine-tuning process.

**Multi-task LoRA Mixture** A LoRA adapter is fine-tuned for a specific downstream task, limiting its utility to that particular application. To enhance the ability of LLMs to handle multiple tasks, two primary approaches are commonly employed. The first approach involves combining datasets from multiple tasks and fine-tuning a single LoRA module on this aggregated dataset. However, Lin et al. (2024b) have identified significant challenges in encapsulating the specialised knowledge required for diverse domains within a single LLM, often leading to suboptimal performance.

The second approach leverages existing LoRA adapters as interchangeable modules that can be directly integrated into a base LLM. Within this strategy, two distinct directions have emerged. The first direction focuses on architectural designs that combine multiple LoRAs using a learnable weighted sum (Huang et al., 2023) or by implementing unified memory pool designs in CUDA kernels (Sheng et al., 2023). However, these frameworks often require continuous few-shot learning or in-context learning for each individual downstream task and necessitate manual assignment of active LoRAs. This manual intervention poses a significant drawback, as it lacks the capability for autonomous selection and dynamic switching of LoRAs during the inference phase.

The second direction involves developing frameworks that enable dynamic fusion of LoRAs. For instance, Xu et al. (2024) introduced MeteoRA, a token-level Mixture-of-Experts (MoE) style multi-task LoRA framework. MeteoRA incorporates a trainable gating mechanism across all attention and MLP layers to automatically select and fuse different LoRAs based on input tokens. While MeteoRA successfully facilitates dynamic management of multiple tasks, the inclusion of a trainable gating module at every attention and MLP layer with token-level routing significantly increases inference

time compared to single LoRA inference. This performance drawback remains substantial even with the development of GPU kernel acceleration methods.

## 3 METHODOLOGY

Our proposed DLP-LoRA framework comprises three key components: a lightweight mini-MLP plugin $\mathcal{C}_{\text{MLP}}$, a base LLM backbone $\mathcal{M}$, and a set of $N$ fine-tuned LoRA modules $L_{\{1...N\}}$ corresponding to different tasks $\mathcal{D}_{\{1...N\}}$, as illustrated in Figure 1. Initially, we train the mini-MLP classifier $\mathcal{C}_{\text{MLP}}$ on these tasks to achieve high task classification accuracy (we evaluate 26 tasks in this work; see Appendix C for details). Once trained, the LLM backbone $\mathcal{M}$ utilises the mini-MLP plugin to dynamically fuse the appropriate fine-tuned LoRAs $L_{\{1...N\}}$ at the sentence level, enabling efficient multi-task learning.

### 3.1 LIGHTWEIGHT MULTI-TASK CLASSIFICATION PLUGIN

Previous methods that perform token-level task classification and routing within the LLM backbone—by injecting a trainable gating network at each attention and MLP layer—are computationally intensive and inefficient during inference (Xu et al., 2024). Observing that most tokens within a sentence typically pertain to the same task, we propose a more efficient sentence-level task detection approach. Specifically, we introduce an off-the-shelf 4-layer mini-MLP plugin $\mathcal{C}_{\text{MLP}}$ that requires training only once on the sentence level for the selected tasks.

Given $N$ distinct tasks $\mathcal{D}_{\{1...N\}}$ and a collection of $M$ sentences $\mathcal{S}_{\{1...M\}} \in \mathcal{D}_n$, our lightweight 4-layer $\mathcal{C}_{\text{MLP}}$ encodes each input sentence $\mathcal{S}_m$ using a specific tokenizer (we utilise the ALBERT tokenizer (Lan, 2019) in this work) and classifies $\mathcal{S}_m$ to the correct task $\mathcal{D}_n$:

$$\mathcal{Y}_n = \mathcal{C}_{\text{MLP}}(\mathcal{S}_m), \quad \text{where} \quad \mathcal{Y}_n \in \mathcal{D}_{\{1...N\}}. \tag{3}$$

### 3.2 DYNAMIC LORA FUSION

Once the $\mathcal{C}_{\text{MLP}}$ classifier is well-trained on the tasks $\mathcal{D}_{\{1...N\}}$, it serves as a plugin to the LLM backbone $\mathcal{M}$ for dynamically fusing multiple LoRAs $L_{\{1...N\}}$ at the sentence level. For the current input sentence $\mathcal{S}_m \in \mathcal{D}_n$, we consider the first token $\text{w}_1$ and the previous contextual history $\mathcal{H}_{\{1...k\}}$. We employ a top-$p$ sampling scheme via $\mathcal{C}_{\text{MLP}}$ to dynamically select the possible LoRAs to fuse, using probability $p$ as the threshold:

$$\mathcal{I}_p = \{\mathcal{Y}_{\{1...R\}} \mid \text{w}_1 \in \mathcal{S}_m, \mathcal{H}_{\{1...k\}}\}, \quad \text{where} \quad \mathcal{Y}_r \geq p. \tag{4}$$

Using the set $\mathcal{I}_p$ for the current sentence $\mathcal{S}_m$, we fuse the selected LoRAs based on normalised weights obtained via a softmax function:

$$\mathcal{W}_m = \text{Softmax}(\mathcal{I}_p) = \{w_1, \ldots, w_R\}. \tag{5}$$

Importantly, the $\mathcal{C}_{\text{MLP}}$ classifier is only activated when the first token $\text{w}_1$ of the current sentence $\mathcal{S}_m$ is generated, leveraging the contextual information $\mathcal{H}_{\{1...k\}}$. This approach significantly accelerates the inference time of $\mathcal{M}$ compared to token-level gating network classification (Xu et al., 2024), as it avoids the overhead of per-token classification.

### 3.3 PARALLEL MULTI-LORA ACCELERATION

Beyond the efficiency gained from sentence-level LoRA sampling and fusion—which avoids the inefficiency of repetitive per-token LoRA classification—a significant advantage of our approach is the ability to fully exploit parallel multi-LoRA acceleration.

Given $N$ fine-tuned LoRAs, we construct two tensors $\mathcal{A} \in \mathbb{R}^{N \times h \times r}$ and $\mathcal{B} \in \mathbb{R}^{N \times r \times d}$, which are allocated contiguously in High Bandwidth Memory (HBM). In contrast to token-level LoRA classification and forward computation—where each token in the batch operates independently, limiting the effectiveness of General Matrix Multiplication (GEMM) optimisations in frameworks like PyTorch—our sentence-level LoRA classification removes the independence constraints among tokens within a sentence. By iterating over all $N$ LoRAs using a hash table stored in HBM, we retrieve the

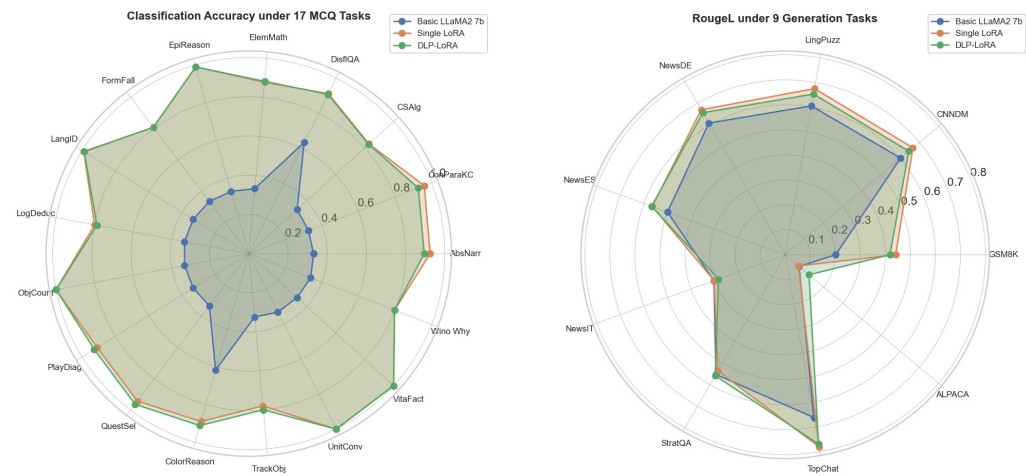

Figure 2: The performance of DLP-LoRA compared to the Basic LLaMA-2 7B and single LoRA baselines across 17 MCQ tasks and 9 QA tasks using accuracy and Rouge-L metrics. See Appendix D for more results using different LLMs backbones.

sampled LoRAs $\mathcal{I}_p$ based on top-$p$ sampling and their corresponding weights $\mathcal{W}_m$. Subsequently, all sampled LoRAs are fused into the original layer-wise weights $\boldsymbol{W}$ of the LLM as follows:

$$\underbrace{[\Delta\boldsymbol{o}_1,\dots,\Delta\boldsymbol{o}_{BM}]}_{B \times M} = \sum_R \boldsymbol{W}^{B \times M \times R}((\underbrace{[\boldsymbol{x}_1,\dots,\boldsymbol{x}_{BMR}]}_{B \times M \times R} \times \underbrace{[\boldsymbol{A}_1,\dots,\boldsymbol{A}_{BMR}]}_{B \times M \times R}) \times \underbrace{[\boldsymbol{B}_1,\dots,\boldsymbol{B}_{BMR}]}_{B \times M \times R})$$

(6)

where $B$ is the batch size, $M$ is the number of sentences, $R$ is the number of sampled LoRAs, and $\boldsymbol{x}$ represents the encoded representation of the first token of each input sentence $\mathcal{S}_m$. Leveraging this parallel multi-LoRA acceleration, our DLP-LoRA achieves an inference time that is on average only 1.24 times slower than single LoRA inference (see Section 4.2 for detailed comparisons).

## 4 EXPERIMENTS

### 4.1 EXPERIMENTAL SETUP

**Datasets.** To comprehensively evaluate our proposed DLP-LoRA framework, we follow the methodology of Xu et al. (2024) and conduct experiments across 26 diverse tasks. These include 17 multiple-choice question (MCQ) datasets covering domains such as mathematical question answering, logical reasoning, language identification, and reading comprehension. Additionally, we assess performance on 9 question-answering (QA) datasets focused on summarisation, machine translation, and open-domain QA. Specifically, we utilise 20 tasks from the BigBench benchmark (Srivastava et al., 2023), 3 machine translation tasks from the News Commentary dataset (Tiedemann, 2012) translating from non-English to English, and 3 generative tasks: GSM8K (Cobbe et al., 2021), CNN/DailyMail (See et al., 2017), and Alpaca (Taori et al., 2023). Detailed descriptions of each dataset are provided in Appendix C.

**LLM Backbones, LoRAs, and Mini-MLP Plugin.** We evaluate DLP-LoRA using four widely adopted LLM backbones: Qwen-2 1.5B and 7B (Yang et al., 2024a), LLaMA-2 7B (Touvron et al., 2023), and LLaMA-3 8B (Dubey et al., 2024). To assess the effectiveness of DLP-LoRA in scenarios with significant model size differences, we also compare the performance of DLP-LoRA based on the Qwen-2 1.5B backbone against the baseline LLaMA-2 13B model without LoRA adaptations, representing a 10x difference in model size.

For the baseline comparisons involving single LoRA modules, we fine-tune a separate LoRA for each task using 900 training samples, randomly selected according to a 9:1 train/test split from each original dataset. The mini-MLP plugin, responsible for task classification, is trained on the

Table 1: The classification accuracy results on 17 MCQ tasks by comparing different basic LLMs backbones, single LoRA baselines and our DLP-LoRA approach. The evaluation results are averaged after running 10 times. The underline indicates the second-best accuracy and the subscript percentage denotes relative accuracy improvement or reduction over each single LoRA baseline.

| Task | Qwen-2 1.5B | | | Qwen-2 7B | | | LLaMA-2 7B | | | LLaMA-3 8B | | |
|---|---|---|---|---|---|---|---|---|---|---|---|---|
| | Basic | LoRA | DLP-LoRA | Basic | LoRA | DLP-LoRA | Basic | LoRA | DLP-LoRA | Basic | LoRA | DLP-LoRA |
| AbsNarr | 33.12 | 89.25 | 89.75 | 27.87 | 93.25 | 92.75 | 33.05 | 92.50 | 89.50 | 86.53 | 97.38 | 97.25 |
| ConParaKC | 24.50 | 100.00 | 93.75 | 24.75 | 99.00 | 94.00 | 32.67 | 96.00 | 92.75 | 84.10 | 98.00 | 95.13 |
| CSAlg | 25.25 | 97.50 | 98.75 | 25.00 | 100.00 | 100.00 | 33.33 | 99.00 | 98.75 | 78.40 | 99.50 | 99.00 |
| DisflQA | 55.59 | 87.57 | 88.10 | 54.44 | 89.63 | 87.98 | 61.80 | 89.03 | 91.17 | 87.96 | 94.42 | 89.97 |
| ElemMath | 25.50 | 81.00 | 81.25 | 25.75 | 85.75 | 86.00 | 32.46 | 78.00 | 80.00 | 88.95 | 90.00 | 90.50 |
| EpiReason | 25.00 | 99.75 | 99.50 | 27.59 | 100.00 | 100.00 | 33.33 | 100.00 | 100.00 | 84.26 | 100.00 | 100.00 |
| FormFall | 25.75 | 100.00 | 100.00 | 25.00 | 100.00 | 100.00 | 33.33 | 100.00 | 100.00 | 83.40 | 100.00 | 100.00 |
| LangID | 27.23 | 77.00 | 77.00 | 25.75 | 89.25 | 88.00 | 33.89 | 79.75 | 79.75 | 76.41 | 95.12 | 94.50 |
| LogDeduc | 35.50 | 84.50 | 80.75 | 25.00 | 89.50 | 90.75 | 33.33 | 83.00 | 82.75 | 93.08 | 96.00 | 96.38 |
| ObjCount | 49.67 | 89.01 | 88.00 | 45.45 | 94.74 | 93.89 | 63.49 | 91.11 | 90.71 | 92.30 | 97.06 | 97.27 |
| PlayDiag | 25.00 | 89.00 | 88.00 | 25.50 | 90.75 | 89.75 | 33.33 | 87.75 | 88.25 | 75.73 | 95.00 | 94.75 |
| QuesSel | 33.52 | 99.00 | 98.00 | 51.11 | 98.00 | 97.00 | 33.00 | 99.00 | 99.00 | 70.41 | 97.00 | 97.00 |
| ColorReason | 25.00 | 79.00 | 78.25 | 25.50 | 87.50 | 87.75 | 33.33 | 80.75 | 80.75 | 82.27 | 95.62 | 96.25 |
| TrackObj | 27.75 | 79.75 | 78.75 | 26.25 | 81.00 | 82.25 | 33.33 | 80.00 | 78.75 | 85.17 | 90.00 | 90.50 |
| UnitConv | 27.11 | 100.00 | 100.00 | 25.00 | 100.00 | 100.00 | 33.33 | 100.00 | 100.00 | 82.67 | 100.00 | 100.00 |
| VitaFact | 32.85 | 94.00 | 92.25 | 30.00 | 96.50 | 95.50 | 33.33 | 90.93 | 92.70 | 79.04 | 96.12 | 95.38 |
| WinoWhy | 43.62 | 94.75 | 96.00 | 30.21 | 91.25 | 93.50 | 33.33 | 94.25 | 96.25 | 88.43 | 96.12 | 96.88 |
| Avg. | 31.88 | 90.65 | 89.89$_{-0.84\%}$ | 30.60 | 93.30 | 92.89$_{-0.44\%}$ | 36.69 | 90.65 | 90.65$_{-0.00\%}$ | 83.48 | 96.31 | 95.93$_{-0.12\%}$ |

Table 2: The BLEU, ROUGE-1 and ROUGE-L results on 9 QA tasks by comparing different basic LLMs backbones, single LoRA baselines and our DLP-LoRA approach. The evaluation results are averaged after running 10 times. The underline indicates the second-best performance and the subscript percentage denotes relative BLEU, ROUGE-1 and ROUGE-L improvement or reduction over each single LoRA baseline.

| Task | Metric | Qwen-2 1.5B | | | Qwen-2 7B | | | LLaMA-2 7B | | | LLaMA-3 8B | | |
|---|---|---|---|---|---|---|---|---|---|---|---|---|---|
| | | Basic | LoRA | DLP-LoRA | Basic | LoRA | DLP-LoRA | Basic | LoRA | DLP-LoRA | Basic | LoRA | DLP-LoRA |
| GSM8K | BLEU | 80.00 | 84.70 | 83.87 | 83.02 | 91.54 | 91.59 | 69.30 | 80.06 | 79.46 | 71.55 | 80.48 | 78.32 |
| | ROUGE-1 | 85.19 | 87.39 | 86.78 | 88.73 | 94.44 | 94.25 | 69.20 | 81.29 | 80.28 | 77.05 | 83.40 | 81.04 |
| | ROUGE-L | 83.27 | 85.96 | 85.16 | 87.54 | 94.28 | 94.08 | 66.12 | 78.00 | 76.89 | 71.65 | 81.00 | 77.83 |
| CNNDM | BLEU | 18.23 | 15.12 | 18.61 | 13.74 | 16.07 | 14.17 | 8.21 | 8.02 | 14.31 | 13.92 | 8.96 | 17.90 |
| | ROUGE-1 | 25.00 | 16.92 | 18.98 | 14.68 | 16.90 | 15.51 | 7.81 | 7.39 | 13.22 | 13.99 | 9.70 | 18.93 |
| | ROUGE-L | 16.94 | 15.83 | 17.19 | 13.48 | 15.40 | 14.03 | 7.30 | 6.95 | 12.45 | 13.25 | 8.80 | 17.76 |
| LingPuzz | BLEU | 39.43 | 43.34 | 42.03 | 40.96 | 57.24 | 56.83 | 44.01 | 58.02 | 56.40 | 36.30 | 64.56 | 65.68 |
| | ROUGE-1 | 25.00 | 29.36 | 26.67 | 23.25 | 47.77 | 46.72 | 23.25 | 45.38 | 43.90 | 24.02 | 58.63 | 58.93 |
| | ROUGE-L | 20.68 | 27.78 | 26.03 | 20.20 | 44.13 | 45.95 | 20.20 | 44.13 | 41.85 | 20.37 | 57.24 | 58.07 |
| NewsDE | BLEU | 62.67 | 64.16 | 64.26 | 61.46 | 63.60 | 68.79 | 66.77 | 69.40 | 67.64 | 62.55 | 68.17 | 58.71 |
| | ROUGE-1 | 60.38 | 66.65 | 63.82 | 60.21 | 64.26 | 68.96 | 61.66 | 67.73 | 65.49 | 61.19 | 70.39 | 68.81 |
| | ROUGE-L | 58.68 | 67.53 | 62.67 | 59.02 | 63.25 | 67.64 | 60.19 | 66.63 | 64.50 | 59.96 | 69.16 | 67.81 |
| NewsES | BLEU | 64.66 | 66.66 | 67.30 | 66.68 | 68.87 | 66.85 | 68.03 | 68.71 | 66.96 | 63.69 | 69.11 | 69.23 |
| | ROUGE-1 | 61.70 | 67.53 | 67.23 | 64.43 | 69.62 | 68.06 | 62.49 | 68.75 | 66.46 | 62.02 | 69.87 | 69.15 |
| | ROUGE-L | 60.61 | 65.70 | 65.69 | 62.73 | 68.68 | 66.99 | 60.49 | 67.50 | 65.31 | 60.34 | 69.16 | 67.83 |
| NewsIT | BLEU | 61.04 | 63.52 | 64.43 | 63.52 | 69.63 | 65.12 | 68.03 | 69.66 | 67.37 | 62.79 | 65.59 | 68.44 |
| | ROUGE-1 | 59.62 | 65.10 | 65.01 | 63.42 | 71.40 | 66.19 | 62.49 | 67.80 | 66.49 | 63.39 | 67.21 | 69.88 |
| | ROUGE-L | 57.74 | 64.38 | 64.37 | 62.11 | 70.77 | 65.85 | 60.49 | 67.00 | 65.63 | 62.50 | 66.52 | 69.29 |
| StratQA | BLEU | 56.69 | 60.67 | 63.25 | 60.44 | 67.75 | 68.02 | 65.11 | 65.58 | 66.51 | 59.26 | 64.34 | 66.22 |
| | ROUGE-1 | 53.28 | 57.86 | 60.99 | 58.13 | 67.28 | 67.72 | 62.08 | 59.87 | 60.08 | 56.83 | 62.82 | 63.45 |
| | ROUGE-L | 49.23 | 54.60 | 56.86 | 54.12 | 64.97 | 65.60 | 60.85 | 56.78 | 56.72 | 52.87 | 60.03 | 60.14 |
| TopChat | BLEU | 30.00 | 32.00 | 29.00 | 25.76 | 33.59 | 34.77 | 59.96 | 33.63 | 33.69 | 32.90 | 35.95 | 29.58 |
| | ROUGE-1 | 30.97 | 31.12 | 29.71 | 26.30 | 33.66 | 35.93 | 53.72 | 32.20 | 30.23 | 31.34 | 35.94 | 30.30 |
| | ROUGE-L | 27.83 | 28.28 | 26.93 | 24.27 | 31.66 | 33.85 | 50.01 | 30.17 | 28.31 | 28.49 | 33.52 | 27.80 |
| ALPACA | BLEU | 62.73 | 62.18 | 66.04 | 62.40 | 63.86 | 63.79 | 37.08 | 64.65 | 66.42 | 63.42 | 64.40 | 63.35 |
| | ROUGE-1 | 60.02 | 57.20 | 63.85 | 52.86 | 61.46 | 61.22 | 32.91 | 59.19 | 61.73 | 58.25 | 61.78 | 61.22 |
| | ROUGE-L | 54.13 | 52.25 | 57.52 | 52.86 | 56.13 | 56.00 | 30.28 | 53.62 | 55.86 | 52.20 | 56.58 | 56.34 |
| Avg. | BLEU | 52.82 | 54.70 | 55.42$_{+1.32\%}$ | 53.11 | 59.13 | 58.83$_{-0.51\%}$ | 53.91 | 57.52 | 57.64$_{+0.21\%}$ | 51.82 | 57.95 | 58.60$_{+1.12\%}$ |
| | ROUGE-1 | 50.56 | 52.99 | 53.67$_{+1.28\%}$ | 50.86 | 58.52 | 58.28$_{-0.41\%}$ | 48.20 | 54.40 | 54.21$_{-0.35\%}$ | 49.79 | 57.75 | 57.96$_{+0.36\%}$ |
| | ROUGE-L | 47.68 | 50.87 | 51.36$_{+0.96\%}$ | 48.53 | 56.82 | 56.66$_{-0.28\%}$ | 45.64 | 52.27 | 51.95$_{-0.61\%}$ | 46.84 | 55.72 | 55.88$_{+0.29\%}$ |

same samples and achieves an average classification accuracy of 98.45%. Notably, the mini-MLP plugin is lightweight, containing only 5 million parameters, and can be trained rapidly—in under 10 minutes—for all 26 tasks. All experiments are conducted on a single custom-upgraded NVIDIA GTX 2080Ti GPU with 22GB of memory.

**Evaluation Metrics and Composite Task Setting.** Given that all 26 tasks can be categorised into MCQ and QA types, we employ accuracy as the evaluation metric for MCQ tasks and BLEU, ROUGE-1, and ROUGE-L scores for QA tasks. To assess multi-task learning capabilities, we create composite task settings by combining the 17 MCQ tasks (Composite-17) and the 9 QA tasks (Composite-9). In all experiments, we report the average results over 10 runs to ensure statistical reliability.

Table 3: Evaluation results for composite-$n$ task, where composite-9 includes all QA tasks, and composite-17 includes all MCQ tasks. In addition, we compare a single LoRA with a higher rank trained on composite-26 task setting. The evaluation results are averaged after running 10 times. The subscript percentage denotes relative accuracy, BLEU, ROUGE-1 and ROUGE-L improvement or reduction over each basic LLMs baseline.

| Model | Method | Composite-$n$ | Acc. (%) ↑ | BLEU ↑ | ROUGE-1 ↑ | ROUGE-L ↑ |
|---|---|---|---|---|---|---|
| **Qwen-2 1.5B** | Basic | 9
17 | -
31.65 | 51.48
- | 48.69
- | 45.72
- |
| | LoRA ($r = 64$) | 26 | 33.23 | 51.46 | 48.86 | 45.90 |
| | DLP-LoRA | 9
17 | -
**90.43** | **56.00**
- | **54.61**
- | **52.27**
- |
| **Qwen-2 7B** | Basic | 9
17 | -
58.59 | 53.25
- | 50.70
- | 48.58
- |
| | LoRA ($r = 64$) | 26 | 59.42 | 53.63 | 51.75 | 48.92 |
| | DLP-LoRA | 9
17 | -
**92.75** | **57.44**
- | **56.84**
- | **54.90**
- |
| **LLaMA-2 7B** | Basic | 9
17 | -
36.29 | 52.32
- | 46.78
- | 44.36
- |
| | LoRA ($r = 64$) | 26 | 37.93 | 52.84 | 46.96 | 45.35 |
| | DLP-LoRA | 9
17 | -
**91.20** | **58.61**
- | **54.70**
- | **52.60**
- |
| **LLaMA-3 8B** | Basic | 9
17 | -
65.44 | 52.00
- | 50.16
- | 47.16
- |
| | LoRA ($r = 64$) | 26 | 65.98 | 52.26 | 50.38 | 47.40 |
| | DLP-LoRA | 9
17 | -
**96.03** | **57.79**
- | **57.45**
- | **55.35**
- |
| **Avg.** | Basic | 9
17 | -
47.99 | 52.26
- | 49.08
- | 46.46
- |
| | LoRA ($r = 64$) | 26 | 49.14 | 52.55 | 49.49 | 46.89 |
| | DLP-LoRA | 9
17 | -
**92.60**$_{+92.95\%}$ | **57.46**$_{+9.95\%}$
- | **55.90**$_{+13.90\%}$
- | **53.78**$_{+15.76\%}$
- |

## 4.2 EXPERIMENTAL RESULTS

**Main Results.** Figure 2 presents the classification accuracy across the 17 MCQ tasks and ROUGE-L scores across the 9 QA tasks, comparing our DLP-LoRA with the baseline LLaMA-2 7B backbone and individually fine-tuned single LoRAs. Our DLP-LoRA not only significantly outperforms the baseline LLaMA-2 7B model but also achieves performance comparable to, and in some cases surpassing, that of the manually loaded single LoRAs on the 17 MCQ tasks. Similar trends are observed for the 9 QA tasks (additional results for other LLM backbones are provided in Appendix D). As shown in Table 1, DLP-LoRA achieves performance within a relative difference of -0.35% in accuracy across the 17 MCQ tasks when compared to the single LoRA models using different LLM backbones. Remarkably, DLP-LoRA consistently outperforms the single LoRA models on the ElemMath and WinoWhy datasets. A similar pattern emerges in Table 2 for the 9 QA tasks, where DLP-LoRA shows relative improvements in BLEU, ROUGE-1, and ROUGE-L scores by averages of 0.54%, 0.22%, and 0.09% across all QA tasks and LLM backbones, respectively. These results demonstrate that DLP-LoRA can match or even exceed the performance of individually fine-tuned single LoRAs by dynamically selecting and fusing multiple LoRAs.

**Multi-task Composite Performance.** We further evaluate DLP-LoRA's capability in multi-task learning under composite task settings by combining the 17 MCQ tasks and the 9 QA tasks. As presented in Table 3, DLP-LoRA significantly enhances performance over the baseline LLM backbones, achieving relative improvements of 92.95% in accuracy for the MCQ composite, and 9.95%, 13.90%, and 15.76% in BLEU, ROUGE-1, and ROUGE-L scores, respectively, for the QA composite. These findings indicate that DLP-LoRA effectively and automatically selects the appropriate LoRAs based on the input prompts within composite tasks, facilitating dynamic multi-task adaptation. A detailed example illustrating how DLP-LoRA selects and fuses multiple LoRAs is provided in Section 4.3.

Table 4: The averaged inference time ratio across 26 datasets by comparing the single LoRA, and DLP-LoRA equiped ALBERT and mini-MLP plugin with the basic LLMs backbones. The subscript percentage denotes relative inference time improvement or reduction of DLP-LoRA over the single LoRA inference.

| Method | Qwen-2 1.5B | Qwen-2 7B | LLaMA-2 7B | LLaMA-3 8B |
|---|---|---|---|---|
| LoRA | 1.15 | 1.00 | 1.05 | 1.00 |
| DLP-LoRA (ALBERT) | $\underline{1.90}_{+65.22\%}$ | $\underline{1.13}_{+13.00\%}$ | $\underline{1.80}_{+71.43\%}$ | $\underline{1.12}_{+12.00\%}$ |
| DLP-LoRA (mini-MLP) | $\underline{1.12}_{-2.61\%}$ | $\underline{1.12}_{+12.00\%}$ | $\underline{1.60}_{+52.38\%}$ | $\underline{1.11}_{+11.00\%}$ |

**Inference Time Efficiency.** We also conduct a comprehensive evaluation of the inference time efficiency of DLP-LoRA and its variants compared to the baseline LLM backbones and single LoRA models. As shown in Table 4, single LoRA models exhibit inference speeds comparable to the baseline LLMs, being only about 1.05 times slower on average. When incorporating ALBERT (11M parameters) as the plugin, DLP-LoRA's inference time ranges from 1.12 to 1.90 times slower than the baseline LLMs, representing a 40.41% increase compared to single LoRA inference. By contrast, using the mini-MLP plugin with 5M parameters, DLP-LoRA achieves faster inference, with only an 18.19% average increase in inference time over single LoRA models across all tasks. These results validate the efficiency of our sentence-level LoRA selection and fusion approach.

## 4.3 CASE STUDY

To illustrate the practical effectiveness of DLP-LoRA, we present a case study in Figure 3 using the LLaMA-3 8B backbone under a composite task setting involving three tasks. For the first input prompt, DLP-LoRA selects two LoRAs—AbsNarr and GSM8K—with probabilities of 50.5% and 49.5%, respectively, using top-$p$ sampling. The AbsNarr dataset involves narratives encapsulating human experiences and wisdom, while GSM8K focuses on practical scenarios requiring general knowledge through mathematical reasoning. The gold standard dataset, StratQA, requires answering general knowledge questions with reasoning steps. DLP-LoRA effectively fuses the AbsNarr and GSM8K LoRAs to generate logical explanations that incorpo-

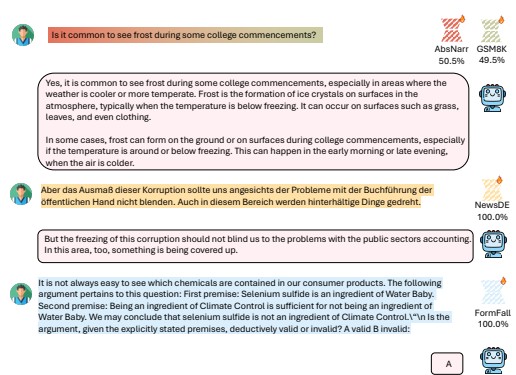

Figure 3: Case study of DLP-LoRA based on LLaMA-3 8B backbone under composite-3 task setting, where the selected LoRAs with corresponding probabilities are demonstrated on the right side.

rate general knowledge about frost weather and commencements. When subsequent questions are input, concatenated with the history, DLP-LoRA continues to successfully select the appropriate LoRAs—NewsDE and FormFall—from the pool of 26 LoRAs stored in high-bandwidth memory (HBM). This case study demonstrates DLP-LoRA's ability to dynamically select and fuse multiple LoRAs to address diverse tasks effectively.

## 5 DISCUSSION

**Limitations of Top-$k$ Selection** Most existing Multi-LoRA or LoRA-MoE methods employ a top-$k$ router to manually determine the fixed number of LoRAs to use for multi-task learning (Li et al., 2024). This manual selection can restrict the model's ability to dynamically select and fuse multiple LoRAs based on the task requirements. In our approach, we utilise top-$p$ selection, which leverages the probabilities assigned by the mini-MLP plugin to each LoRA, using a threshold $p$. This allows DLP-LoRA to adaptively decide both the number and combination of LoRAs to fuse for different tasks, enhancing flexibility and performance.

**Can a Smaller LLM with DLP-LoRA Outperform a Larger LLM Backbone?** Our evaluations of DLP-LoRA across various LLM backbones ranging from 1.5B to 8B parameters under

Table 5: The comparison between smaller Qwen-2 1.5B equipped with DLP-LoRA and basic LLaMA-2 13B backbone under composite-26 task setting. The subscript percentage denotes relative inference time, each evaluation metric improvement or reduction of Qwen-2 1.5B + DLP-LoRA over the LLaMA-2 13B.

| Model | Inference Time (s) ↓ | Acc. (%) ↑ | BLEU ↑ | ROUGE-1 ↑ | ROUGE-L ↑ |
|---|---|---|---|---|---|
| LLaMA-2 13B | 4672.73 | 45.54 | 19.89 | 17.99 | 17.25 |
| Qwen-2 1.5B + DLP-LoRA | $425.03_{-90.90\%}$ | $82.68_{+81.55\%}$ | $20.59_{+3.52\%}$ | $39.57_{+119.96\%}$ | $38.84_{+125.16\%}$ |

composite task settings prompted us to investigate whether a smaller LLM backbone equipped with DLP-LoRA can outperform a larger, unadapted LLM backbone. As shown in Table 5, the Qwen-2 1.5B model equipped with DLP-LoRA reduces inference time by over 90% compared to the LLaMA-2 13B backbone when processing a mixture of 26 tasks. Moreover, it achieves significant improvements in accuracy, ROUGE-1, and ROUGE-L scores by 81%, 119%, and 125%, respectively. These findings suggest that smaller LLMs augmented with DLP-LoRA have the potential to match or even surpass the performance of much larger models (with over eight times more parameters) across diverse tasks. This is particularly beneficial for deployment on devices with limited computational resources, such as mobile devices.

**Inference Time of Multi-LoRA Loading at Scale** By avoiding inefficient and repetitive token-level LoRA classification, our method fully leverages PyTorch's General Matrix Multiplication (GEMM) operations for parallel multi-LoRA acceleration. We conducted an ablation study to assess how the inference time scales with the increasing number of LoRAs, using the LLaMA-3 8B backbone as a reference. As illustrated in Table 6, even as the number of LoRAs increases, the inference time ratio re-

Table 6: The inference time ratio compared between different numbers of LoRAs and the basic LLaMA-3 8B. # Params denote the percentage of LoRAs' parameters over the LLaMA-3 8B.

| Num. of LoRA | # Params | Inference Time Ratio |
|---|---|---|
| 50 | 0.043% | 1.76 |
| 100 | 0.085% | 1.83 |

mains within 2x of the baseline LLaMA-3 8B model. Additionally, the combined parameters of all LoRAs constitute less than 0.1% of the 8B parameters in the LLaMA-3 backbone. These results demonstrate that our approach scales efficiently with the number of LoRAs without incurring significant computational overhead, maintaining practical inference times even at scale.

**Efficiency comparison with different dynamic LoRAs baselines** We further compare our DLP-LoRA with different dynamic LoRAs baselines in order to evaluate the DLP-LoRA's efficiency at inference speed and memory usage. We fine-tuned 7 different LoRAs based on selected 7 datasets including ARC (Clark et al., 2018), HellaSwag (Zellers et al., 2019), MMLU (Hendrycks et al., 2020), TruthfulQA (Lin et al., 2022),

Table 7: The inference time and memory consuming ratio compared with different dynamic LoRAs baselines based on LLaMA-2 7B. The subscript percentage denotes relative inference time and memory usage improvement of different LoRAs baselines over the LLaMA-2 7B backbone.

| Method | Decoding latency ratio | Peak Memory ratio |
|---|---|---|
| LLaMA2-7B | 1.00 | 1.00 |
| MOLA (Gao et al., 2024) | $10.54_{+954\%}$ | $2.04_{+104\%}$ |
| PESC (Wu et al., 2024a) | $3.54_{+254\%}$ | $\mathbf{1.02}_{+2\%}$ |
| MoRAL (Yang et al., 2024b) | $3.58_{+258\%}$ | $\mathbf{1.02}_{+2\%}$ |
| LoRA-Switch (Kong et al., 2024) | $\mathbf{1.29}_{+29\%}$ | $1.07_{+7\%}$ |
| DLP-LoRA (7 LoRAs) | $\mathbf{1.20}_{+20\%}$ | $\mathbf{1.00}_{+0\%}$ |

WinoGrande (Sakaguchi et al., 2021), ScienceQA (Lu et al., 2022), CommonsenseQA (Talmor et al., 2019), and OpenbookQA (Mihaylov et al., 2018). Then we compare DLP-LoRA with different baselines on the ShareGPT dataset (Wang et al., 2023) [1] following LoRA-Swich (Kong et al., 2024). As shown in Table 7, it is evident that DLP-LoRA stands out in both speed and memory efficiency. Even when handling seven tasks, DLP-LoRA completes inference tasks quickly with minimal additional memory costs, demonstrating a significant advantage over other methods. With our DLP plugin method, switching to a different LoRA requires only retraining a small 5M mini-MLP, resulting in minimal computational overhead. This simplifies the implementation of new MoE plugins. Furthermore, DLP-LoRA maintains strong performance even with a large number of LoRAs, a scenario

---

[1] Since LoRA-Switch did not descript how many LoRAs are utilised during inference for ShareGPT dataset, we assume that all 7 LoRAs based on the original work are equipped and we can regard this as the lower-bound of DLP-LoRA.

where other methods often struggle in Table 7. This robustness is advantageous for applications requiring multiple LoRAs. Additionally, DLP-LoRA effectively minimizes the increase in parameters. For example, using LLaMA-3 8B with 100 MoE dynamic LoRAs, a typical gating method would add approximately 26M parameters, calculated as hidden size × LoRA types × hidden layers × 2 (accounting for query and value matrices). In contrast, DLP-LoRA only adjusts the final linear layer, keeping the total increase to around 5M parameters. This suggests that LoRA fine-tuning can enable LLMs to enhance their capabilities across various domains simultaneously when equipped with sufficient LoRAs.

## 6 RELATED WORK

In the area of multi-task learning with LoRA, two primary research directions have emerged beyond the straightforward approach of fine-tuning a single LoRA on a combined dataset of multiple tasks (Lin et al., 2024b). The first direction focuses on developing libraries or frameworks to reuse and integrate existing LoRAs, while the second aims to design router networks based on MoEs to dynamically fuse multiple LoRAs.

**Multiple LoRA Architectures**    Several works have proposed frameworks for combining and managing multiple LoRAs. Huang et al. (2023) introduced LoRAHub, a framework that combines existing fine-tuned LoRAs using a learnable weighted sum, allowing for more flexible adaptation across tasks. S-LoRA (Sheng et al., 2023) emphasises unified memory pool design to manage dynamic LoRA weights with varying ranks and key-value cache tensors for CUDA kernels, enhancing computational efficiency. Additionally, Model-Based Clustering (MBC) (Ostapenko et al., 2024) employs clustering techniques to group tasks based on the similarity of their LoRA parameters, facilitating better parameter sharing and task generalization.

**Mixture-of-Experts with Multiple LoRAs**    Another line of research integrates Mixture-of-Experts mechanisms to control and fuse multiple LoRAs dynamically. In these approaches, multiple LoRAs are fine-tuned and injected into the model's MLP layers, with a router network determining which LoRA to activate for a given task. Examples include LoRAMoE (Dou et al., 2024), PHATGOOSE (Muqeeth et al., 2024), MoLE (Wu et al., 2024b), and LoRA-Switch (Kong et al., 2024). Some methods extend this fusion to both MLP and attention layers, such as MixLoRA (Li et al., 2024) and Mixture of Adaptations (MoA) (Feng et al., 2024), enabling more comprehensive adaptation across model components.

Furthermore, token-level routing strategies have been proposed to enhance the granularity of LoRA selection. MeteoRA (Xu et al., 2024) introduces a token-level MoE-style multi-task LoRA framework with trainable gating mechanisms across all attention and MLP layers, allowing for dynamic selection and fusion of different LoRAs based on input tokens. Similarly, AdaMoE (Zeng et al., 2024) presents an adaptive MoE approach that leverages token-level routing within transformer models to improve performance across diverse tasks. Apart from the token-level gating mechanism for multiple LoRAs, some existing works also proposed sentence-level routing, for instance Polytropon (Ponti et al., 2023) and FLix (Lin et al., 2024a). However, Flix mainly focuses on multilingual task settings and Polytropon mainly explores the encoder-decoder architecture. It is unclear whether those works can maintain superior inference efficiency when loading high volumes of LoRAs across different tasks.

## 7 CONCLUSION

We introduced DLP-LoRA, a dynamic and lightweight plugin that employs a mini-MLP module with only 5 million parameters to dynamically fuse multiple LoRAs at the sentence level using top-$p$ sampling strategies. Our comprehensive evaluation across 17 MCQ tasks and 9 QA tasks demonstrates that DLP-LoRA not only closely matches the performance of individually fine-tuned single LoRAs but also surpasses them on certain tasks, all while incurring less than twice the inference time. Through detailed discussions and ablation studies, we have shown that DLP-LoRA effectively balances performance and efficiency in multi-task learning, making it a practical solution for dynamic multi-task adaptation in LLMs.

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

## A   LIMITATIONS

Our evaluation of DLP-LoRA was primarily conducted on LLM backbones ranging from 1.5 billion to 8 billion parameters, constrained by the computational limitations of our GPU resources. Consequently, we were unable to assess the performance of DLP-LoRA on larger models such as Qwen-2.5 32B (Hui et al., 2024) and LLaMA-3.1 70B (Dubey et al., 2024), which may exhibit different behaviors and performance characteristics. Additionally, when composite tasks include a higher proportion of MCQ datasets, DLP-LoRA tends to assign higher probabilities to the specific MCQ LoRA, potentially limiting its ability to effectively fuse and utilize QA LoRAs. This tendency might restrict the diversity of generated outputs and the fusion capabilities of DLP-LoRA across a broader range of tasks.

## B   BROADER IMPACTS

The lightweight design of DLP-LoRA, featuring a mini-MLP with only 5 million parameters, offers significant flexibility and efficiency, making it suitable for deployment on smaller devices with limited computational resources. Moreover, DLP-LoRA facilitates easy integration of new LoRAs corresponding to additional tasks without necessitating further fine-tuning of the entire model. This capability enhances the accessibility and adaptability of LLMs in various applications, promoting broader utilisation in resource-constrained environments.

## C   DETAILS ABOUT 26 TASKS AND DATASETS

Table 8 includes detailed descriptions of each dataset's name, keywords, main content and corresponding evaluation metrics. These 26 tasks include diverse topics, such as mathematical QA, logical reasoning, language identification, reading comprehension, summarisation, machine translation, and open-domain QA.

## D   EVALUATION RESULTS BASED ON DIFFERENT LLMS BACKBONES

We demonstrate more radar charts to show more results for each LLM backbone. Figure 4, 5 and 6 demonstrate that DLP-LoRA significantly outperforms the basic LLM backbones under 17 MCQ datasets, and DLP-LoRA also outperforms the basic LLaMA-3 8B a lot across 17 MCQ datasets in Figure 3. In addition, we can find that DLP-LoRA achieves comparable performance of single LoRA mode based on different LLM backbones from 1.5B to 8B under 9 QA tasks.

Table 8: Details about the 26 selected tasks following Xu et al. (2024).

| Task Name | Keywords | Description | Evaluation Metrics |
|---|---|---|---|
| abstract_narrative_understanding (AbsNarr) | narrative understanding, multiple choice | Given a narrative, choose the most related proverb. | Accuracy |
| alpaca (ALPACA) | instruction-tuning | Write appropriate answers according to instructions. | BLEU, ROUGE |
| cnn_dailymail (CNNDM) | summarization | Given news articles, write the summarization. | ROUGE |
| contextual_parametric_knowledge_conflicts (ConParaKC) | contextual question-answering, multiple choice | Answer questions given the contextual information. | Accuracy |
| cs_algorithms (CSAlg) | algorithms, numerical response | Solve two common computer-science tasks. | Accuracy |
| disfl_qa (DisflQA) | contextual question-answering, reading comprehension | Pick the correct answer span from the context given the disfluent question. | Accuracy |
| elementary_math_qa (ElemMath) | mathematics | Answer multiple choice mathematical word problems. | Accuracy |
| epistemic_reasoning (EpiReason) | logical reasoning, multiple choice | Determine whether one sentence entails the next. | Accuracy |
| formal_fallacies_syllogisms_negation (FormFall) | logical reasoning, multiple choice, | Distinguish deductively valid arguments from formal fallacies. | Accuracy |
| gsm8k (GSM8K) | mathematics | Solve the grade school math word problems. | Accuracy |
| language_identification (LangID) | multilingual, multiple choice | Given a sentence, select the correct language. | Accuracy |
| linguistics_puzzles (LingPuzz) | logical reasoning, linguistics | Solve Rosetta Stone-style linguistics puzzles. | BLEU, ROUGE |
| logical_deduction (LogDeduc) | logical reasoning, multiple choice | Deduce the order of a sequence of objects. | Accuracy |
| news_commentary_de (NewsDE) | multilingual, translation | Translate German sentences into English. | BLEU |
| news_commentary_es (NewsES) | multilingual, translation | Translate Spanish sentences into English. | BLEU |
| news_commentary_it (NewsIT) | multilingual, translation | Translate Italian sentences into English. | BLEU |
| object_counting (ObjCount) | logical reasoning | Questions that involve enumerating objects and asking the model to count them. | Accuracy |
| play_dialog_same_or_different (PlayDiag) | reading comprehension, multiple choice | Determine if nearby lines in a Shakespeare play were spoken by the same individual. | Accuracy |
| question_selection (QuestSel) | reading comprehension, multiple choice | Given an answer along with its context, select the most appropriate question which has the given answer as its answer. | Accuracy |
| reasoning_about_colored_objects (ColorReason) | reading comprehension, logical reasoning, multiple choice | Answer extremely simple questions about the colors of objects on a surface. | Accuracy |
| strategyqa (StratQA) | logical reasoning, context-free question answering | Answer questions in which the required reasoning steps are implicit in the question. | BLEU, ROUGE, Accuracy |
| topical_chat (TopChat) | free response | Open-domain response generation. | BLEU, ROUGE |
| tracking_shuffled_objects (TrackObj) | logical reasoning, multiple choice | Determine the final positions given initial positions and a description of a sequence of swaps. | Accuracy |
| unit_conversion (UnitConv) | contextual question-answering, mathematics, multiple choice | Perform various tasks relating to units, including identification and conversion. | Accuracy |
| vitaminc_fact_verification (VitaFact) | truthfulness, reading comprehension, multiple choice | Identify whether a claim is True or False based on the given context. | Accuracy |
| winowhy (WinoWhy) | causal reasoning, multiple choice | Evaluate the reasoning in answering Winograd Schema Challenge questions. | Accuracy |

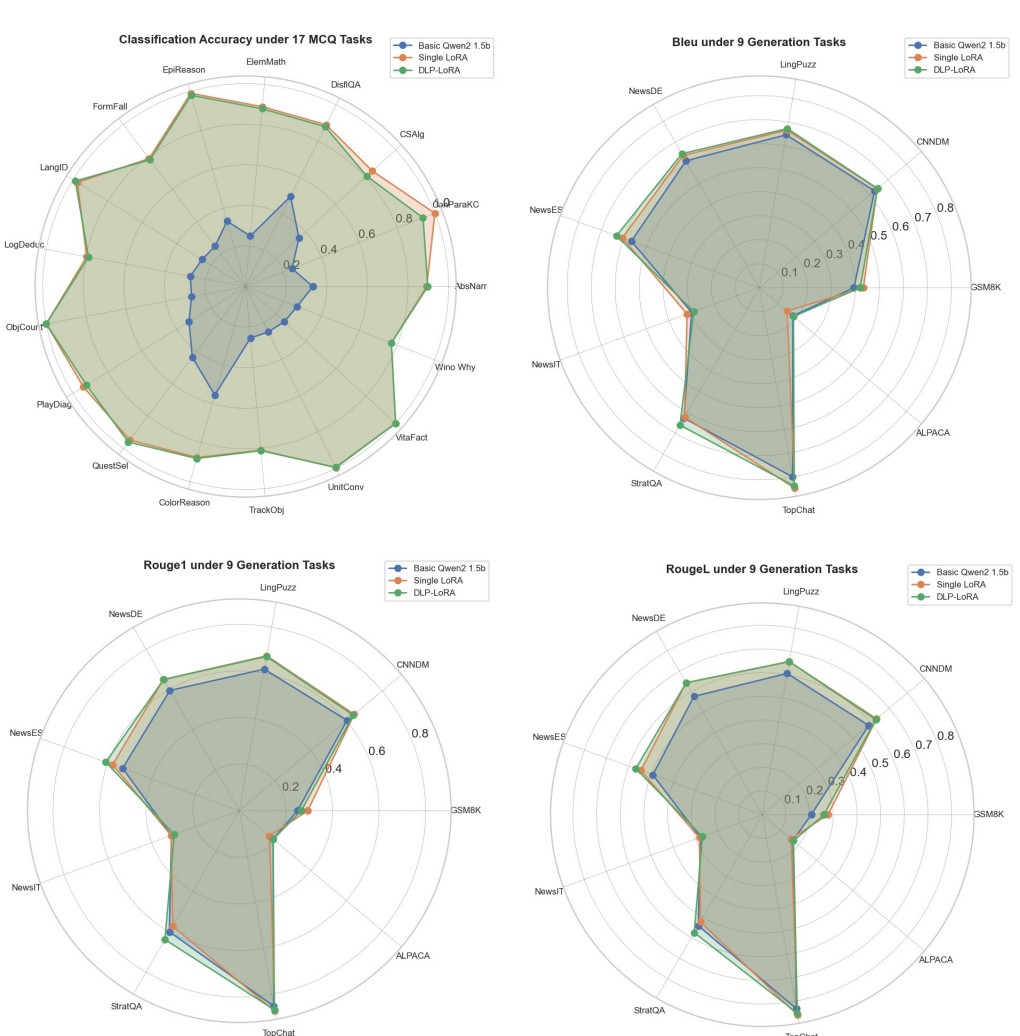

Figure 4: Radar chart of Qwen-2 1.5B across 17 MCQ and 9 QA tasks.

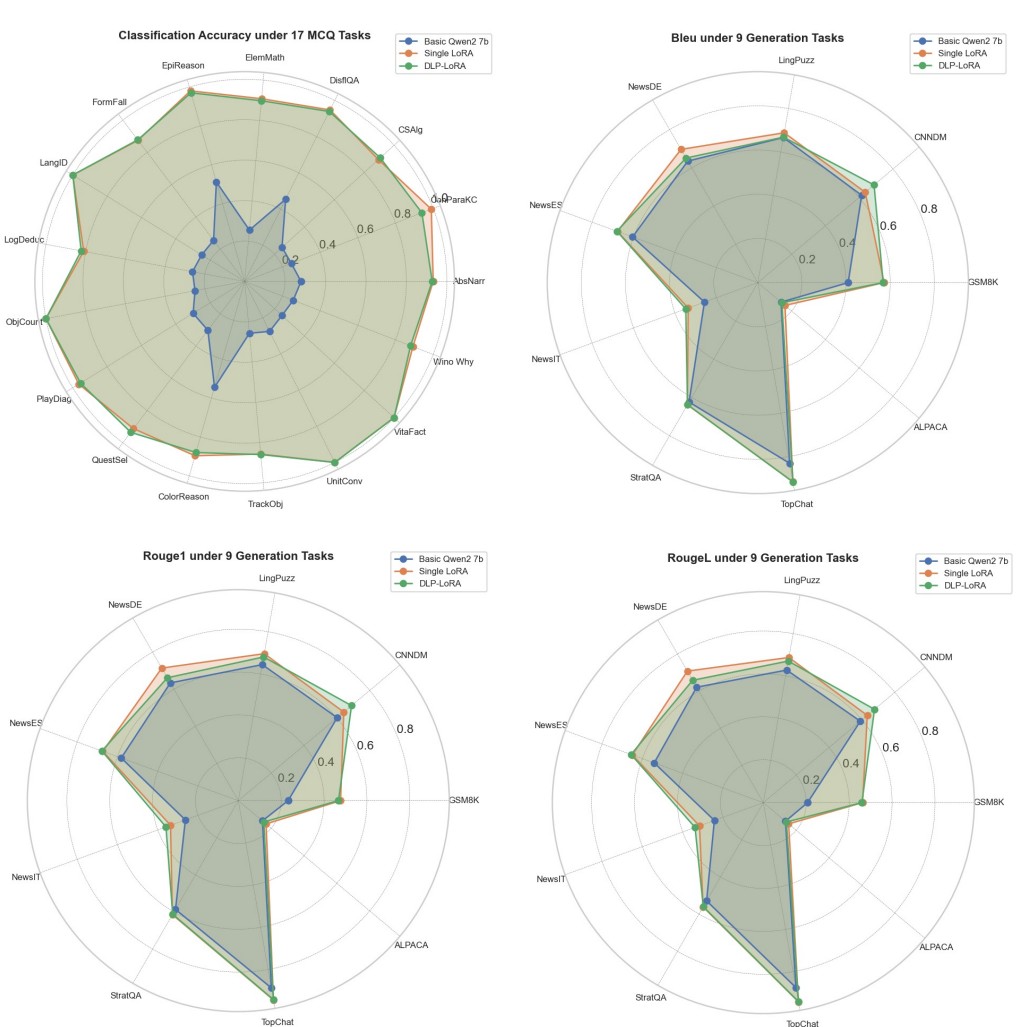

Figure 5: Radar chart of Qwen-2 7B across 17 MCQ and 9 QA tasks.

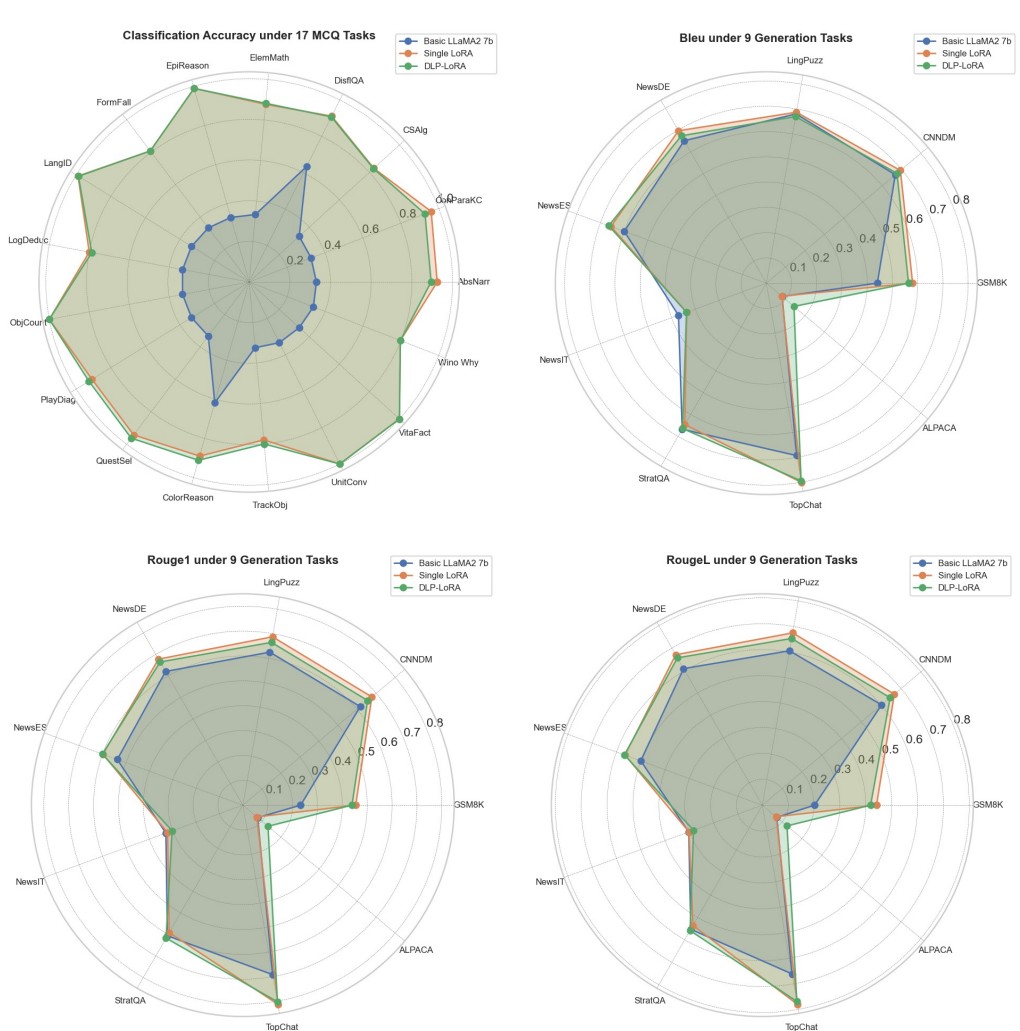

Figure 6: Radar chart of LLaMA-2 7B across 17 MCQ and 9 QA tasks.

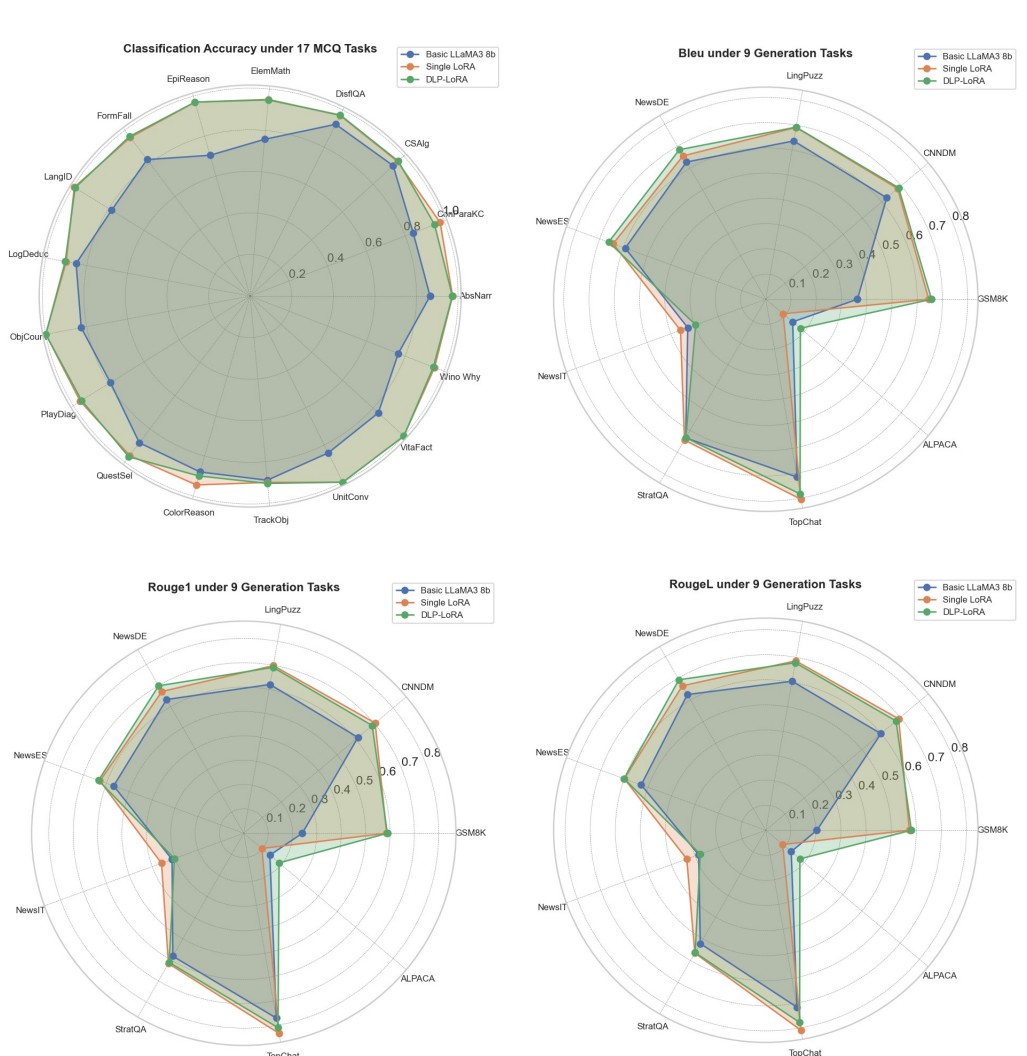

Figure 7: Radar chart of LLaMA-3 8B across 17 MCQ and 9 QA tasks.

