# OpenReview forum: "DLP-LoRA: Efficient Task-Specific LoRA Fusion with a Dynamic, Lightweight Plugin for Large Language Models"
_ICLR.cc/2025/Conference — Submitted to ICLR 2025_

### Official Review · Reviewer_fMGx · 2024-10-21

**Soundness:** 2
**Presentation:** 2
**Contribution:** 2
**Rating:** 3
**Confidence:** 3

**Summary:**

- The study proposes a dynamic lightweight plugin to fuse multiple LoRA adapters that enable the adaptation of language models to specific downstream tasks and new data domains.
- The proposed approach involves a “mini-MLP plugin” that combines the weights of multiple LoRA modules based on the contextual information provided by the language model input.
- The authors DLP method aims to improve efficiency over token-level LoRA mixture approaches by utilizing sentence-level representations. The performance benchmark indicates that their DLP-LoRA method achieves similar performance to single task-specific LoRA modules.

**Strengths:**

- The evaluation across 26 diverse tasks including multiple choice questions, question answering, summarization, translation is quite extensive. They include ablation studies evaluating inference time and contrast model size and performance results.
- The authors clearly describe the benefits of their proposed DLP-LoRA method.
- the performance evaluation includes multiple medium-sized decoder-only language models with Llama and Qwen

**Weaknesses:**

- The differences to related methods are not clear or not specifically evaluated in regards to their performance or efficiency contribution. How does the approach differ from methodologically from i.e. PHATGOOSE [1], or LoRAMoE [2] and how does the performance differ?
- The authors list various mixture of expert methods in their related works and introduction, but only include single LoRAs and the base model in their benchmark results.
- Some claims remain unsupported by the empirical results presented in the study. The authors mention that additional fine-tuning for new tasks is not required in their DLP framework, yet there is no performance benchmark on unseen tasks. Furthermore, they describe the framework as lightweight but lack the parameter and space complexity comparison with related LoRA mixture approaches.
- The description of their proposed method lacks details regarding the training of the LoRA modules that are combined to perform the downstream tasks.

References:
- [1] Muqeeth, M., Liu, H., Liu, Y., & Raffel, C. (2024). Learning to Route Among Specialized Experts for Zero-Shot Generalization. arXiv [Cs.LG].
- [2] Shihan Dou, Enyu Zhou, Yan Liu, Songyang Gao, Wei Shen, Limao Xiong, Yuhao Zhou, Xiao Wang, Zhiheng Xi, Xiaoran Fan, Shiliang Pu, Jiang Zhu, Rui Zheng, Tao Gui, Qi Zhang, and Xuanjing Huang. 2024. LoRAMoE: Alleviating World Knowledge Forgetting in Large Language Models via MoE-Style Plugin. In Proceedings of the 62nd Annual Meeting of the Association for Computational Linguistics (Volume 1: Long Papers), pages 1932–1945, Bangkok, Thailand. Association for Computational Linguistics.

**Questions:**

- Are the single LoRAs included in the selection of different LoRA modules DLP is combining?
- Can you elaborate why the performance of multiple LoRA modules combined through your DLP method, which include the single task-specific LoRA module, is similar or worse than the single-LoRA module?
- The inference time comparisons in Table 4 are not explained in the caption. Metric and reference time is missing.
- Section 5 lists selecting adapters based on their relevance vs. fixed k as a key limitation of related work. Do you have empirical results that support this claim for your DLP method?
- Can you elaborate on the differences between the base model performances in Table 1? Specifically, Llama 3 8b seems to perform significantly better than Llama 2, and both Qwen models.

---

> ### Author Response · Authors · 2024-11-21
> **Replies by Authors**
>
> We are grateful for your detailed feedback and address your points below:
>
> 1.**Comparison with Different Baselines Regarding Performance on Multitask Datasets**
>
>    We apologize for any misunderstanding about our contribution. Our work primarily introduces the lightweight sentence-level mini-MLP plugin that efficiently serves multiple LoRAs. As evidenced in Table 3, our method handles up to 100 LoRAs with minimal impact on inference time compared to the base LLaMA-3 8B model. We also highlight the efficiency in inference time over manually loading individual LoRAs, as shown in Table 7.
>
> 2. **Inference Efficiency Comparison Between Different Baselines**
>
>    Recognizing the importance of this comparison, we have included relevant experiments in Table 7. Our DLP-LoRA demonstrates superior performance in decoding latency ratio and peak memory ratio when using LLaMA-2 7B as the backbone. The advantage stems from our sentence-level routing approach, which contrasts with token-level routing methods that increase inference time due to per-token processing.
>
> 3. **LoRA Modules Training for Downstream Tasks**
>
>    We adhere to the standard LoRA fine-tuning procedure for each task. The key difference in our approach is that we select and fuse different LoRAs at the sentence level, meaning all tokens within a sentence share the same fused LoRAs.
>
> 4. **Claims About Additional Fine-tuning of Other Methods for New Tasks**
>
>    To clarify:
>
>    Methods using token-level routing, such as S-LoRA and MeteoRA, require the entire base model and all associated LoRAs to be loaded during fine-tuning for a new task, which is resource-intensive. Our approach allows for independent fine-tuning of the mini-MLP (only 5 million parameters) and a single LoRA for the new task, making it feasible on less powerful hardware.
>
> 5. **Combining Single LoRAs for Different LoRA Selection**
>
>    As detailed in Figure 1 (right side) and Section 3.2, we use the mini-MLP to assign probabilities to each LoRA. We then fuse the A and B weight matrices of the selected LoRAs based on these normalized probabilities, applying the final fused weights to downstream tasks.
>
> 6. **Explanation of Performance Difference Between Classification and Generation Tasks**
>
>    We acknowledge that our initial explanation was insufficient. In our updated version, we expand on this point:
>
>    Due to the limited output space in classification tasks (typically a single token), the performance may be slightly impacted when fusing multiple LoRAs, as irrelevant information can be introduced. In contrast, generation tasks benefit from the fusion of multiple LoRAs, as the longer outputs allow the model to produce more nuanced and contextually appropriate responses.
>
> 7. **Clarification in Table 4 Caption**
>
>    We apologize for any confusion. In Table 4, the inference time ratio is calculated by dividing the inference time of the standard LoRA by that of DLP-LoRA (mini-MLP). This ratio allows for consistent comparison across different hardware setups, facilitating reproducibility.
>
> 8. **Explanation of Limitations of Top-k Sampling**
>
>    Thank you for this insightful question. Fixed top-k sampling lacks flexibility in multi-task settings. For classification tasks, a high top-k may introduce noise from irrelevant LoRAs, degrading performance. Conversely, for QA and generation tasks, a low top-k may omit relevant LoRAs, hindering performance. Our method employs a probability threshold  p  for selecting and fusing LoRAs, providing a more adaptable and task-appropriate selection mechanism.
>
> 9. **Explanation for Superior Performance of LLaMA-3 8B Compared to Other LLM Backbones**
>
>    The datasets in Table 1 are predominantly from the BIG-Bench collection. According to the official Hugging Face repository (https://huggingface.co/meta-llama/Meta-Llama-3-8B), LLaMA-3 8B outperforms LLaMA-2 7B on these tasks. While direct comparisons between LLaMA-3 8B and Qwen models on BIG-Bench are not officially available, our observations indicate that Qwen models underperform on classification tasks in this benchmark compared to LLaMA-3 8B. This discrepancy may result from differences in pretraining datasets and strategies between Qwen and LLaMA-3 models.

---

> > ### Author Response · Authors · 2024-11-25
> > **Replies to Reviewer fMGx**
> >
> > Dear Reviewer fMGx, we explained each raised question during our rebuttal. We are looking forward to your further comments and valuable feedback regarding our responses.

---

> > > ### Comment · Reviewer_fMGx · 2024-11-25
> > > **Response to Authors**
> > >
> > > Thank you for the additional clarifications and improvements included in the manuscript. I believe that my current evaluation remains appropriate, as the paper requires a thorough revision to incorporate the additional information.

---

### Official Review · Reviewer_ohza · 2024-10-30

**Soundness:** 2
**Presentation:** 2
**Contribution:** 2
**Rating:** 3
**Confidence:** 4

**Summary:**

This paper focuses on achieving a more efficient dynamic fusion of multiple LoRA experts in multi-task adaptation for LLMs. To this end, the authors propose DLP LoRA, which operates as a sentence-level Mixture-of-Experts MoE gating mechanism. Specifically, a pre-trained task classifier is used to obtain the probability distribution of the input sentence across various tasks, and then the gating mechanism fuses selected LoRAs filtered by a fixed probability threshold. Additionally, the paper presents a parallel CUDA acceleration strategy to improve inference efficiency. Experiments are conducted on 26 tasks adaptation, achieving performance comparable to single-task LoRA.

**Strengths:**

1. The paper provides a deeper exploration into previous methods for dynamically fusing multiple LoRA experts and attempts incremental improvements.

2. The topic investigated in this paper has considerable application value.

**Weaknesses:**

1. The proposed DLP-LoRA is largely incremental and lacks sufficient novelty.

2. The experimental evaluation is unconvincing: 1）The baselines in Tables 1, 2, and 3 are weak and insufficient, and the authors should consider introducing relevant methods from this field for comparison; 2）The paper emphasizes the proposed methodʼs efficiency advantages, yet lacks quantitative experiments on efficiency compared to single LoRA inference and previous methods.

3. There are some issues in the writing that reflect a lack of rigor: 1）The paper incorrectly refers to top-p sampling; to my knowledge, top-p sampling restricts the candidate set based on cumulative probability thresholds, whereas the paper independently filters LoRA experts below a fixed probability threshold, which is inconsistent; 2）The introduction claims that previous methods require additional fine-tuning when tasks change, yet DLP-LoRA seems unable to solve this problem either.

**Questions:**

1. The paper repeatedly emphasizes that the proposed method achieves less than twice the inference time of single LoRA inference; Is there quantitative experimental data to support this claim?

2. Why does Table 5 compare the LLaMA-2 13B with a smaller, fine-tuned LLM? From an inference speed perspective, a smaller LLM outperforming a larger LLM is expected; from a performance perspective, a fine-tuned model outperforming an un-fine-tuned model is also expected. Is this a fair comparison?

---

> ### Author Response · Authors · 2024-11-21
> **Replies by Authors**
>
> Thank you for your valuable feedback. We address your concerns below:
>
> 1. **Comparison with Different Baselines Regarding Performance on Multitask Datasets**
>
>    We apologize for any confusion regarding our main contribution. Our work primarily focuses on the lightweight sentence-level mini-MLP plugin that efficiently serves multiple LoRAs. As demonstrated in Table 6, our method can handle up to 100 LoRAs with less than twice the inference time of the base LLaMA-3 8B model. We emphasize the efficiency gains in inference time compared to manually loading single LoRAs, as presented in Table 7 of the updated version.
>
> 2. **Inference Efficiency Comparison Between Different Baselines**
>
>    Recognizing the importance of this comparison, we have added quantitative experiments and compared our DLP-LoRA with several baselines in Table 7. The results show that our DLP-LoRA outperforms state-of-the-art baselines in decoding latency ratio and peak memory ratio using LLaMA-2 7B as the backbone. This improvement arises because token-level routing methods require per-token routing, leading to increased inference time. Our sentence-level routing mechanism, facilitated by the mini-MLP plugin, significantly reduces inference time by deciding which LoRAs to fuse at the sentence level.
>
> 3. **Incorrect Usage of Top-p Sampling Concept**
>
>    Thank you for bringing this to our attention. We have corrected the usage of the top-p sampling term in our revised version to accurately reflect our method.
>
> 4. **Claims About Additional Fine-tuning of Other Methods for New Tasks**
>
>    We apologize for any lack of clarity. To clarify:
>
>    Token-level routing methods like S-LoRA and MeteoRA require loading the base model and all relevant LoRAs simultaneously for fine-tuning new tasks. For large models (e.g., 70B parameters), this necessitates substantial computational resources, such as multiple high-end GPUs. In contrast, our DLP-LoRA only needs to fine-tune the mini-MLP (approximately 5 million parameters) and a single LoRA for the new task. This process is resource-efficient and can be conducted on CPUs or GPUs with less memory.
>
> 5. **Table 5 Comparison Between Qwen-2 1.5B with 26 DLP-LoRAs and LLaMA-2 13B**
>
>    The purpose of this comparison is to demonstrate that our method maintains strong performance across 26 tasks while offering fast inference speeds when loading multiple DLP-LoRAs simultaneously. For comparisons based on the same backbone LLMs, please refer to Tables 1, 2, 3, and 7, which showcase the superior capabilities of our DLP-LoRA.

---

> > ### Comment · Reviewer_ohza · 2024-11-24
> > **Official Comment by Reviewer ohza**
> >
> > I appreciate the rebutal of authors. However, I will maintain my rating since the new explanations and results are not sufficiently convincing.

---

> > > ### Author Response · Authors · 2024-11-24
> > > **Replies to Reviewer ohza**
> > >
> > > Thanks for your comments. Could the reviewer ohza point out which explanations and results that are not sufficiently convincing? We could give more explanations regarding the specific point.

---

### Official Review · Reviewer_4fCq · 2024-11-03

**Soundness:** 1
**Presentation:** 2
**Contribution:** 1
**Rating:** 3
**Confidence:** 3

**Summary:**

This paper proposes a method to dynamically fuse pre-trained task-specific LoRA modules. The idea is to train a sentence-level router for different tasks, and simply use that for routing and fusing different LoRA modules. The method is evaluated on both classification and language generation tasks such as QA.

**Strengths:**

1. the paper is addressing an important problem in efficiently re-using expert modules. The proposed method is more efficient since it uses sentence-level routing, and it is evaluated on various classification and language generation tasks.
2. the paper also includes discussions on inference efficiency, which is very relevant to the practical usage of the method.

**Weaknesses:**

1. the paper lacks important baselines. the routing module is trained jointly on N tasks, so the method assumes that one has access to all the task-specific training data for each LoRA module. Therefore, the authors should compare to a joint multitask training LoRA setting where the LoRA module is optimized on all task data. Note that the LoRA for joint multitask training should have a larger rank so that the number of tunable parameters is equivalent to tuning individual LoRA modules for each task.
2. the results in table 1 shows that the proposed method is worse than baselines for classification, but it's better for generation tasks in table 2. However, there is not enough explanation or intuition on why that's the case.
3. there are some highly relevant work on compositional LoRA with sentence-level routing that's not cited: https://aclanthology.org/2023.eacl-main.49.pdf, https://arxiv.org/abs/2402.17934

**Questions:**

1. how does your method compares to tuning a LoRA module with a higher rank on all multitask training data?

---

> ### Author Response · Authors · 2024-11-21
> **Replies by Authors**
>
> Thank you for your thoughtful comments. We address your points below:
>
> 1. **Comparison with a Single LoRA Fine-tuned on All Multitask Data with Higher Rank**
>
>    We appreciate your suggestion. We have included this new baseline in our updated version of Table 3. The experiments reveal that the single LoRA fine-tuned jointly on all tasks, even with a higher rank, performs worse and suffers from catastrophic forgetting compared to our DLP-LoRA. This indicates that a single LoRA's capacity, even when increased, is insufficient to effectively handle all 26 tasks.
>
> 2. **Explanation of the Performance Difference Between Classification and Generation Tasks**
>
>    We acknowledge that our initial explanation was insufficient. In our updated version, we provide a more detailed discussion:
>
>    Classification tasks typically require the model to output a single token representing the answer choice, offering limited linguistic variability. Consequently, performance on classification tasks may be slightly affected when fusing multiple LoRAs due to potential interference. In contrast, generation tasks involve producing longer and more varied outputs, allowing DLP-LoRA to better leverage the fused LoRAs to generate meaningful and coherent responses, as illustrated in Figure 3.
>
> 3. **Missing Related Work**
>
>    We apologize for omitting relevant references. We have updated our paper to include these works and have added discussions in the related work section to contextualize our contributions within existing literature.

---

> ### Author Response · Authors · 2024-11-25
> **Replies to Reviewer 4fCq**
>
> Dear Reviewer 4fCq, we explained each raised question during our rebuttal. We are looking forward to your further comments and valuable feedback regarding our responses.

---

### Author Response · Authors · 2024-11-21
**General Comments**

We sincerely appreciate all the reviewers' constructive feedback and their recognition of our proposed method's contributions, particularly in inference efficiency (Reviewers 4fCq and fMGx), comprehensive evaluation across multiple tasks (Reviewer fMGx), and significant application value (Reviewer ohza).

Based on your insightful comments, we would like to address some common concerns:

- **Baseline Comparison Regarding Inference Efficiency**: We acknowledge that our original submission lacked an inference time comparison across different baselines. Following your suggestions, we have added this experiment of Table 7 to our updated version. Our results show that DLP-LoRA significantly outperforms several dynamic LoRA methods in terms of decoding latency ratio and peak memory ratio. This demonstrates the effectiveness of our sentence-level routing mechanism and dynamic LoRA fusion based on a probability threshold.

- **Main Contribution of Our Work**: As highlighted in our paper's title, introduction, and experiments, our primary contribution lies in proposing a lightweight plugin that effectively fuses multiple LoRAs. This makes our method adaptable to LLMs of varying sizes and capable of handling a high volume of LoRA loadings. We focus on comparing with different LoRA baselines to showcase the superior inference efficiency of DLP-LoRA. Additionally, we compare with basic LLM backbones and standard LoRAs to demonstrate that DLP-LoRA maintains comparable performance across 26 diverse tasks under both single-task fine-tuning and composite task settings.

We hope that our responses address your concerns and clarify the contributions and advantages of our proposed method. Thank you again for your valuable feedback and we are looking forward to your replies.

---

### Meta-Review · Area_Chair_zxZv · 2024-12-19

**Metareview:**

All reviewers agree that the paper addresses an important problem and the methodology looks promising. But there is a consensus among the reviewers that the comparison has been done only with the simple baselines (in terms of performance) and not with the more sophisticated ones (few has been use for comparison for inference time but not accuracy). Adding stronger baselines for performance comparison can make this paper's argument stronger.

**Additional Comments On Reviewer Discussion:**

None

---

### Decision · Program_Chairs · 2025-01-22

Reject